# SARS-CoV-2, COVID-19, and Reproduction: Effects on Fertility, Pregnancy, and Neonatal Life

**DOI:** 10.3390/biomedicines10081775

**Published:** 2022-07-22

**Authors:** Julien Harb, Nour Debs, Mohamad Rima, Yingliang Wu, Zhijian Cao, Hervé Kovacic, Ziad Fajloun, Jean-Marc Sabatier

**Affiliations:** 1Faculty of Medicine and Medical Sciences, University of Balamand, Dekouene Campus, Sin El Fil 55251, Lebanon; julienharb0408@gmail.com (J.H.); nour.debs1@hotmail.com (N.D.); 2Laboratory of Applied Biotechnology (LBA3B), Azm Center for Research in Biotechnology and Its Applications, EDST, Lebanese University, Tripoli 1300, Lebanon; mohamad.rima@hotmail.com; 3StarkAge Therapeutics, Campus de l’Institut Pasteur de Lille, 59000 Lille, France; 4State Key Laboratory of Virology, Modern Virology Research Center, College of Life Sciences, Wuhan University, Wuhan 430072, China; ylwu@whu.edu.cn (Y.W.); zjcao@whu.edu.cn (Z.C.); 5Aix-Marseille Université, CNRS UMR 7051, INP, Inst Neurophysiopathol, 13385 Marseille, France; herve.kovacic@univ-amu.fr; 6Department of Biology, Faculty of Sciences 3, Lebanese University, Campus Michel Slayman Ras Maska, Tripoli 1352, Lebanon

**Keywords:** SARS-CoV-2, COVID-19, ACE2, RAS, fertility, reproduction, neonatal life

## Abstract

Since its discovery in Wuhan, China, severe acute respiratory syndrome coronavirus 2 (SARS-CoV-2) has spread over the world, having a huge impact on people’s lives and health. The respiratory system is often targeted in people with the coronavirus disease 2019 (COVID-19). The virus can also infect many organs and tissues in the body, including the reproductive system. The consequences of the SARS-CoV-2 infection on fertility and pregnancy in hosts are poorly documented. Available data on other coronaviruses, such as severe acute respiratory syndrome (SARS-CoV) and Middle Eastern Respiratory Syndrome (MERS-CoV) coronaviruses, identified pregnant women as a vulnerable group with increased pregnancy-related complications. COVID-19 was also shown to impact pregnancy, which can be seen in either the mother or the fetus. Pregnant women more likely require COVID-19 intensive care treatment than non-pregnant women, and they are susceptible to giving birth prematurely and having their newborns admitted to the neonatal intensive care unit. Angiotensin converting enzyme 2 (ACE2), a key player of the ubiquitous renin-angiotensin system (RAS), is the principal host cellular receptor for SARS-CoV-2 spike protein. ACE2 is involved in the regulation of both male and female reproductive systems, suggesting that SARS-CoV-2 infection and associated RAS dysfunction could affect reproduction. Herein, we review the current knowledge about COVID-19 consequences on male and female fertility, pregnant women, and their fetuses. Furthermore, we describe the effects of COVID-19 vaccination on reproduction.

## 1. Introduction

The novel coronavirus disease 2019 (COVID-19) pandemic, caused by the severe acute respiratory syndrome coronavirus 2 (SARS-CoV-2) viral infection, is a significant, exponentially developing global public health emergency, with new abnormalities being diagnosed and reported on a daily basis [1]. The pandemic touched the entire globe and overwhelmed the medical system [2]. The viral infection shares some epidemiological and clinical features with other coronaviruses, such as severe acute respiratory syndrome (SARS-CoV) and Middle Eastern Respiratory Syndrome (MERS-CoV) (reviewed in [3]). COVID-19 can range from asymptomatic cases, to moderate flu-like symptoms, to severe respiratory illness. The main symptoms of the SARS-CoV-2 infection disease include a dry cough, dyspnea, and fever. Fatigue, musculoskeletal discomfort, headaches, gastrointestinal problems, and a loss of smell and taste are also well-documented [4,5,6]. More studies are now investigating the effects of a SARS-CoV-2 infection on systems other than the respiratory system [7]. Among these, whether the coronavirus can harm the male and female reproductive systems is currently being considered.

Angiotensin-converting enzyme 2 (ACE2) acts as a cellular attachment site to the SARS-CoV-2 spike protein which anchors the virus to the target cells [8]. ACE2 is expressed on several different organs or tissues and is an important component of Renin-Angiotensin System (RAS). Angiotensin-2 (AngII), a product of the cleavage of angiotensin-1 (AngI) by ACE, acts as a potent vasoconstrictor, pro-inflammatory, and pro-fibrotic [9]. AngII can further be cleaved by ACE2 to form the peptide Ang1-7, which counteracts the activity of AngII and has vasodilatory, anti-inflammatory, and anti-fibrotic effects [10]. The balance between these two faces of RAS is therefore assured by ACE2 (For review see [11]). However, SARS-CoV-2 invasion and cellular internalization lead to the down-regulation of membrane-bound ACE2 and increase serum ACE2, which leads to Ang1-7 depletion and an unopposed AngII activity [9] (Figure 1A). Since the RAS is known to be of great importance in regulating different physiological processes (such as vasoconstriction, inflammation, angiogenesis, oxidative stress, and apoptosis) [5], the complications following the SARS-CoV-2 infection are likely due to RAS impairment [12,13,14]. ACE2 can be found at the surface of many cell types, including respiratory epithelial cells, cardiac fibroblasts, cardiomyocytes, endothelial cells, vascular smooth muscle cells (VSMCs), kidneys, gut, the central nervous system (CNS), and the reproductive system [15]. This ubiquitous expression of ACE2 makes different organs susceptible to SARS-CoV-2 infection and explain the multiple-organ damage seen with COVID-19. Notably, the expression of RAS components in both the male and female reproductive systems indicates that they are susceptible to SARS-CoV-2 infection (Figure 1B).

In this review, we summarize the literature reporting the effects of COVID-19 on the male and female reproductive systems. Whether the viral infection affects both men and women’s fertility, and how it impacts pregnancy will be also discussed. We will also address whether the COVID-19 vaccines have any effect on the reproductive systems.

## 2. COVID-19 Effect on Fertility

### 2.1. Does COVID-19 Affect Male Fertility? What Is the Possible Role of RAS?

The interaction of the SARS-CoV-2 viral spike protein with angiotensin converting enzyme 2 (ACE2) on cells co-expressing ACE2 and the cellular transmembrane protease serine 2 (TMPRSS2) has been identified as the SARS-CoV-2 virus’s cellular entry mechanism. Since the testes express ACE2 receptors, researchers are investigating the effects of COVID-19 on male fertility [16]. Endocrinologically, the hypothalamic–pituitary–gonadal (HPG) axis connects the brain and the testes. The production of gonadotropins and testosterone, as well as the HPG feedback loop, are responsible for this connection. The effects of COVID-19 on the hypothalamic–pituitary–gonadal axis are still under investigation; however, aberrant gonadotropin levels have been reported in COVID-19 patients [17].

Maintaining a normal testosterone production requires a healthy HPG axis. Several studies reported that COVID-19 influences testicular hormone production. In fact, researchers discovered high luteinizing hormone (LH) levels and reduced testosterone to LH ratios in COVID-19 patients. These findings were linked to systemic inflammation in the patients examined [18]. Moreover, when compared to more moderate cases, 12.9 percent of patients who died or had severe COVID-19 had reduced total and free testosterone and raised LH [19].

Researchers related circulating gonadotropin levels in COVID-19 patients to neuropathology. Neuroimaging of a single patient indicated hyperintense signals, which could indicate hypothalamic abnormalities as well as an expanded pituitary gland [20]. Even though these data are preliminary, they suggest that hypothalamic perturbation in COVID-19 patients may disrupt gonadotropin release regulation, resulting in a drop in testosterone levels.

Furthermore, it has been proven that SARS-CoV-2 can cross the blood–brain barrier and infect ACE2-expressing cells, causing neuroinflammation [17,21]. Normal physiologic activities like temperature regulation and hormone balance can be disrupted by inflammation [20,22]. Fever is the body’s reaction to systemic inflammation and one of the most reported symptoms of COVID-19 [23]. A temperature higher than 39 °C for more than three days has been linked to a considerable drop in semen concentration and motility [24]. Therefore, the possible effect of COVID-19 on male fertility can be an indirect consequence of associated fever.

The blood–testes barrier offers the testicles special immunity. Inflammation, both systemic and local, can enhance permeability and allow immune cells to invade [25]. A SARS-CoV-2 infection generates a proinflammatory response in the body and can trigger cell pyroptosis, a programmed cell death associated with the production of proinflammatory cytokines. Inflammatory cytokines induce immune cell recruitment, which might lead to a cytokine storm and uncontrolled systemic inflammation affecting various organ systems [26]. Inflammatory markers such as interleukine-6 (IL-6), IL-8, and tumor necrosis factor-alpha (TNF-α) have been detected in semen samples from individuals recovering or suffering from COVID-19 [27,28]. Inflammatory cytokines and oxidative stress have both been shown to harm testes’ biological components [29]. In fact, oxidative stress damages Leydig cells, impairing testosterone synthesis and spermatogenesis [30]. Therefore, the testicular damage reported in COVID-19 patients has been linked to oxidative stress as a possible cause. For example, in an autopsy investigation of COVID-19 patients, a statistically significant increase in reactive oxygen species (ROS) and a reduction in glutathione disulfide (GSH) levels [31].

Furthermore, the membrane-bound ACE2, a significant component of the RAS, is the human gate that allows the SARS-CoV-2 virus to enter host cells [32,33]. Since RAS appears to have various effects on male fertility, it could be possible that COVID-19 consequences on male fertility is linked to RAS impairment. In fact, in human and mammalian animal models, typical components of the RAS have been identified in the testis and epididymis [34,35,36] (Figure 2A). In humans, ACE2 expression is high in the spermatogonia, Leydig, and Sertoli cells of adult human testis in the testes [37,38]. In addition, an abnormal expression of genes implicated in mitochondrial function and testicular steroidogenesis was observed in knockout mammalian models of RAS components [34,39]. Together, the RAS and its ACE2 receptor play an important role in male reproduction by regulating steroidogenesis, testosterone production, and spermatogenesis in the testis in human males (Figure 2B). Since COVID-19 impairs these pathways, it is not surprising that male fertility is affected after SARS-CoV-2 infection. In addition, since ACE2 is expressed on endothelial cells, the SARS-CoV2 infection of these cells can lead to endothelial dysfunction and inflammation affecting male fertility (inducing erectile dysfunction) [40].

### 2.2. The Effect of COVID-19 on Female Fertility

Women that were infected by SARS-CoV-2 reported changes in menstrual cycle frequency and regularity, altered menstrual duration and volume, worsening premenstrual syndrome, and increased dysmenorrhea. Therefore, questions were raised about the impact of COVID-19 on female reproduction after the viral infection.

The abundance of RAS components in the female reproductive system suggests its vulnerability to SARS-CoV-2 infection. ACE2 is widely expressed in the ovary, uterus, vagina, and placenta [15]. ACE2 controls follicular development and ovulation, as well as luteal angiogenesis and degeneration, along with endometrial tissue alterations and embryo development [15]. Other components of RAS are abundant in the female reproductive system such as the Ang1-7 that is expressed in theca-interstitial cells. The presence of ACE2 and Ang1-7 in all phases of follicular development suggests that they may play an important role in fertility [15,36]. Based on these evidence, SARS-CoV-2 infection may disrupt female fertility by damaging ovarian tissue, granulosa cells, and endometrial epithelial cells [41]. By downregulating ACE2 levels, the infection results in higher levels of AngII, which has been linked to proinflammatory, profibrotic, and proapoptotic effects. Consequently, this could influence ovarian function and lead to an increase in ovarian oxidative stress [42]. Of note, SARS-CoV-2 was not identified in vaginal fluid and cervical exfoliated cells, suggesting that the lower female genital tract may not be a transmission route for SARS-CoV-2 [43]. These findings were contradicted in another study in which SARS-CoV-2 was detected using vaginal RT-PCR in some patients [44]. Whether there is a link between viral load and the detection threshold of the virus vaginal levels should be investigated further. Taken together, the findings suggest that the female reproductive system, which expresses ACE2, is vulnerable to SARS-CoV-2 infection, and thus fertility could be affected.

The dynamic expression of RAS in the stromal and epithelial cells of the endometrium during the cycle could explain the changes in the menstrual cycle observed during SARS-CoV-2 infection [15,45,46]. In fact, the RAS is effective for controlling menstrual cycles, enabling blood vessel renewal, and triggering menstruation [15]. The balanced expression of the stimulatory factor AngII and the inhibitory factor Ang1-7 regulates these processes [15]. Since SARS-CoV-2 infection affects RAS, complications touching RAS-regulated physiological systems are possible.

SARS-CoV-2 encodes proteins that can activate the NOD-, LRR-, and pyrin domain-containing protein 3 (NLRP3) inflammasome assembly [47,48]. In fact, one of the first defenses against viral infections is the inflammasome, which is a key player of the innate immunity. When NLRP3 is activated, it attracts Caspase-1, which boosts the expression of interleukins IL-1β and IL-18 [49,50]. Since women with a history of recurrent miscarriages have higher levels of NLRP3 and proinflammatory cytokines in their endometrium [51], it is possible that SARS-CoV-2-associated inflammation affects female fertility.

## 3. The Effect of COVID-19 in Pregnant Women

According to the World Health Organization (WHO, Geneva, Switzerland), pregnant women do not seem to have higher risk of getting a SARS-CoV-19 infection, and they do not show an increased risk of mortality when infected [52]. However, a study showed that most infected pregnant women required hospitalization compared to non-pregnant women [53]. These findings showed that pregnancy confers substantial additional risk of morbidity.

Increased risk of serious outcomes of COVID-19 have been linked to pregnant women who are older, overweight, or have preexisting medical conditions (especially hypertension and diabetes). Out of various maternal characteristics evaluated (including maternal age, gestational age at delivery, gravidity, nulliparity, multiparity, and medical comorbidities), both gestational age at delivery and medical comorbidity showed a statistically significant difference between SARS-CoV-2-negative and SARS-CoV-2-positive pregnant women [54]. However, none of the obstetric complications including anemia, gestational diabetes mellitus, pregnancy-induced hypertension, intrahepatic cholestasis, antepartum hemorrhage, and postpartum hemorrhage showed a statistically significant difference between SARS-CoV-2-positive and SARS-CoV-2-negative pregnant women admitted for delivery [54].

The clinical manifestation of COVID-19 in pregnant women included common symptoms such as fever, cough, myalgia, diarrhea, dyspnea, headache, and chest tightness. Some cases of pregnant women were asymptomatic at the time of admission, whereas others developed severe pneumonia, therefore requiring mechanical ventilation and admission into the intensive care unit (ICU). Mortality cases of COVID-19-positive pregnant women were reported due to severe pneumonia and multiple organ dysfunction [55]. In addition, comparing SARS-CoV-2-positive to SARS-CoV-2-negative pregnant women showed that mild COVID-19 was associated with preeclampsia, preterm birth, and stillbirth. However, severe COVID-19 lead to preeclampsia, preterm birth, gestational diabetes, and low birth weight [56]. These findings suggest that the complications observed in pregnant women could be linked to the severity of the viral infection.

COVID-19 was also linked to increased complications in pregnant women, such as coagulation and respiratory systems (Figure 3). In the following sections, we focus on the perturbation of these two systems during COVID-19, while focusing on pregnancy and the RAS.

### 3.1. Pregnancy, COVID-19, and Coagulopathy

The RAS has an important role in the pathophysiology of coagulopathy in COVID-19 patients. Many reported thromboembolic complications were seen in SARS-CoV-2 positive patients [57]. As such, 29.4% of ICU patients with COVID-19 within a large New-York City health system had a thrombotic event (13.6% venous and 18.6% arterial), whereas 11.5% of non-ICU patients had thrombotic events (3.6% venous and 8.4% arterial) [58]. Another study of hospitalized patients with COVID-19 in China found that 46% developed lower extremity deep venous thrombosis [59].

Since pregnancy is a hypercoagulable state (pregnancy-induced hypercoagulability), COVID-19 can also affect the coagulation cascade in pregnant patients. In pregnancy, there is an increase in thrombin production and in intravascular inflammation that serve as an adaptive mechanism to prevent post-partum bleeding. Several prothrombotic factors such as factors VII, VIII, X, XII, von Willebrand factor and fibrinogen are increased, whereas protein S decreases, in addition to altered fibrinolysis [60]. In addition, during pregnancy, there is an overexpression of many RAS components, such as Ang1-7, which has a vasodilator role [61]. Contrarily, the ACE is decreased during normal pregnancy [62], suggesting a perturbation of the RAS. Together, these findings highlight the possible emergence of coagulopathies in COVID-19 pregnant patients due to an accentuated downregulation of the ACE.

Thus, the hypercoagulable state in both pregnancy and COVID-19 patients may have synergistic risk factors for thrombosis in pregnant women with COVID-19. This conclusion is further supported by the fact that in patients with symptomatic COVID-19, the levels of D-dimer and C-reactive protein (CRP) were about 2.5 and 6 times higher, respectively, compared to pregnant women without SARS-CoV-2 infection [63]. Moreover, a case report describes the first maternal death of a 29-year woman of Pakistani origin at 29 weeks of pregnancy with COVID-19 due to a large pulmonary embolism and basilar artery embolism [64].

### 3.2. Complications of COVID-19 on the Pulmonary System of Pregnant Women

During a healthy pregnancy, several physiological changes occur in the respiratory tract through biomedical and mechanical pathways. Both estrogen and progesterone increase during pregnancy. Estrogen upregulates progesterone receptors within the hypothalamus and medulla, the central neuronal respiratory-related areas, and further increases these receptors’ sensitivity. High progesterone increases both oxygen consumption and tidal volume, leading to an increase in minute ventilation which, in turn, increases the arterial partial pressure of oxygen PO_2_ and decreases the arterial partial pressure of carbon dioxide [65]. The enlarging uterus acts mechanically in displacing the diaphragm superiorly and altering the thoracic configuration. Although there is a 30–40% increase in tidal volume, there is a reduction in functional residual capacity (FRC) and expiratory reserve volume (ERV) [65,66]. Thus, the altered total lung capacity (TLC) during pregnancy may raise women’s susceptibility to pneumonia and respiratory distress syndrome following a SARS-CoV-2 infection.

Furthermore, the AngII/Ang1-7 imbalance during COVID-19 leads to increased vascular permeability, leading to further recruitment of neutrophils into the lung parenchyma [9]. Neutrophil accumulation can lead to alveolar epithelial cell loss through its prooxidative role and the development of acute respiratory distress syndrome (ARDS) [67]. Therefore, pregnant women required extensive follow up and monitoring since severe infection and pulmonary deterioration lead to preterm birth in many reported cases [5]. That was the case of a COVID-19-positive pregnant woman, at 34 weeks, who developed severe ARDS after 4 days of fever and dyspnea. Respiratory failure on the fourth day of admission required an emergency cesarean section. Furthermore, mechanical ventilation was initiated after delivery due to severe distress [68].

On the other hand, a meta-analysis conducted on SARS-CoV-2-positive pregnant women indicated that most women had only a mild form of the disease, and the recovery rate was estimated to be 99.9% [69]. No severe cases of COVID-19 pneumonia were reported. In fact, pneumonia was reported as mild or moderate in 78% of total cases, and 31% were asymptomatic [69].

## 4. The Debate about Vertical Transmission of SARS-CoV-2

Whether vertical transmission occurs during SARS-CoV-2 infection is still under debate, and the risk of fetal infection has not yet been established. Several studies aimed to gather statistical evidence of possible vertical transmission and determine whether the virus impacts the normal development of the fetus. In a normal pregnancy, the placenta acts as a barrier and prevents fetal infection from several microorganisms. Both syncytiotrophoblasts and cytotrophoblasts act as a barrier to infections through complex architecture and innate immune mechanisms [70]. Other immunological defenses are also present in the decidua, including maternal natural killer (NK) cells, decidual macrophages, and T cells, which provide further immunity against pathogens [71].

To assess whether the vertical transmission of SARS-CoV-2 is possible, it is vital to check whether the virus can cross the placenta. As previously mentioned, ACE2 is the main receptor of the virus along with transmembrane serine protease 2 (TMPRSS2), a protease that is essential for SARS-CoV-2 entry and replication in the cells [72]. In several studies, ACE2 expression was detected in syncytiotrophoblasts in COVID-19-positive pregnant women and controls, whereas TMPRSS2 expression was absent in the two groups [72]. Since ACE2 is required for viral entry and replication, syncytiotrophoblasts could be vulnerable to SARS-CoV-2. However, the absence of TMPRSS2, a protease that is also ‘key’ for viral entry makes the hypothesis of viral transmission through this pathway less likely [73], although an TMPRSS2-independent endosomal pathway of SARS-CoV-2 entry into targets cells, via a furin- or cathepsin-based spike protein cleavage (spike protein priming), does exist.

Neonatal COVID-19 status was examined in several reports. According to the American Academy of Pediatrics (AAP) Perinatal COVID-19 Registry, about 2% of infants from more than 3000 deliveries tested positive within 96 h of birth from mothers who tested positive for SARS-CoV-2 around the delivery time. Of 18 newborns with positive tested mothers, 15 tested negative, 2 had unclear results on day of life 0 but then turned out negative when repeated on day 1, and the remaining newborn showed an indeterminate test result that was considered to be negative [74]. In several other studies, vertical transmission was not detected since all neonates tested negative [69,75], creating serious confusion about vertical transmission. Of note, there were no differences in birth weight and presence of asphyxia in neonates when compared between COVID-19-positive and COVID-19-negative mothers [75].

Other studies reported positive neonatal cases and raised the suspicion of vertical transmission [76]. Although the real-time reverse transcriptase-polymerase chain reaction (RT-PCR) for SARS-CoV-2 nucleic acid was negative, several studies assessed the presence of antibodies (Immunoglobulin G (IgG) and Immunoglobulin M (IgM)) in neonates as a possible indicator for intrauterine infection transmission. In one study, six COVID-19-positive mothers had elevated serum IgG and IgM levels. Moreover, 3 of the infants had elevated IgG and elevated IgM. The remaining 3 infants had elevated IgG but normal IgM levels [77]. Similar results showing elevated antibody and cytokine levels (with negative SARS-CoV-2 RT-PCR) obtained 2 h after the birth of a healthy neonate from a COVID-19-positive mother having elevated antibody levels [78]. The reliability of antibodies to assess the possible vertical transmission remains under question, especially given that maternal IgG pass to the fetus via the placenta, which may lead to false positive results. Contrarily, in normal conditions, IgM are less likely to cross the placenta [79], except in the presence of infection, in which mother-to-fetus IgM transfer is increased [79].

## 5. Complications of Neonates Born to SARS-CoV-2 Infected Mothers

As discussed earlier, most of the statistical studies have shown that neonates born to infected mothers had negative RT-PCR results and were asymptomatic. Furthermore, the possibility of vertical transmission occurring is not well defined. However, possible neonatal comorbidities that might occur due to maternal SARS-CoV-2 infection are well documented. In fact, two neonates born to previously healthy mothers, having positive nasopharyngeal swab for SARS-CoV-2, had negative nasopharyngeal swabs for SARS-CoV-2. However, the first baby developed a low-grade fever with abdominal distention and lymphopenia, and the second neonate developed lymphopenia and mild pneumonia [80].

Another study also revealed the effect of maternal COVID-19 infection on 10 newborns with negative RT-PCR results. The newborns showed several symptoms including fever, thrombocytopenia, dyspnea, rapid heart rate, and vomiting. Five neonates were cured and discharged, four remained in the hospital but were stable, and 1 died due to various factors including shock induced by viraemia, multiple organ failure, and disseminated intravascular coagulation [81].

## 6. Vaccination Effect

Multiple COVID-19 vaccines are being developed, licensed, and manufactured quickly due to the severity of the disease. Among these, mRNA vaccines are seen as great options because of their distinctive features. However, certain serious adverse effects have been documented following their administration, raising concerns about the vaccines’ safety and efficacy [82]. Although various fertility associations have stated that COVID-19 mRNA vaccines are unlikely to influence fertility, the existing research is relatively limited, which is one of the causes for vaccine apprehension among the public, particularly among pregnant women. Since the vaccination produces a native-like conformation of the spike protein that can interact with ACE2, it is imaginable that the vaccine impairs RAS and therefore affects fertility. Given the important role of RAS in spermatogenesis (discussed above), the overactivation of AngII and deficiency in Ang1-7, which counteracts the deleterious effects of AngII, could affect male fertility. Further studies should be conducted to investigate this hypothesis.

Both of the COVID-19 mRNA vaccines that have been granted emergency use authorization (mRNA-1273 and BNT162b2) show the potential to promote Th1 immunity and stimulate interferon^+^ CD8^+^ T-cell responses in males and non-pregnant women [83]. Given how important a proper balance of Th1/Th2 immunity is for good obstetric outcomes, these findings led to questions about whether the vaccine’s effect on the cellular immune system could pose a risk to pregnancy. Several studies found that vaccinated gravidas and the general population have similarly low rates of pregnancy problems and unfavorable obstetric outcomes like miscarriage and premature birth [84,85]. In addition, no differences were reported in neonatal complications (such as newborn respiratory complications) between vaccinated and unvaccinated groups [86].

Furthermore, maternal Abs, whether produced after infection or vaccination, may protect neonates from infection, reducing pregnant women’s reluctance to be vaccinated [87,88,89]. In fact, IgG against the spike protein (both S1 and S2) and receptor-binding domain (RBD) of the spike protein of the virus are produced in response to the COVID-19 vaccine, whereas IgG against the spike protein (both S1 and S2), RBD of the spike protein, and other viral proteins are produced following COVID-19 infection. [90]. The antibodies generated in response to COVID-19 vaccination are passed on to the fetus during pregnancy. Blood antibodies from the mother and the fetus were found to be nearly identical [91]. Antibodies to IgG were discovered in 98.5% of babies born to moms who had received two doses of the Pfizer-BioNTech vaccine. On the other hand, 43.6% of neonates whose mothers had received one dose of the Pfizer-BioNTech vaccination developed COVID-19-specific IgG antibodies in their blood [92].

Even though COVID-19 vaccination has been shown to be more or less effective in up to 90% of cases, only about three-quarters of non-pregnant women agreed to get vaccinated, compared to around 50% of pregnant women [93]. In both the Moderna and Pfizer-BioNTech vaccines, the most common consequence in pregnant mothers is, reportedly, injection site discomfort [94].

The incidence of systemic adverse events of Moderna and Pfizer-BioNTech vaccines increased after the second dose of immunization [85]. The most reported systemic adverse effects included fatigue, headaches, shivering, malaise, rash, and vomiting. The majority of these were only transient, and only a few lasted more than three days. The frequency of such systemic adverse events was substantially higher after the second dose than it was after the first dose. The Moderna vaccination group had more participants with these systemic adverse effects than the Pfizer-BioNTech group in terms of numbers.

When compared to unvaccinated expecting mothers, vaccination has apparently no effect on gestation or delivery. No significant changes in the prevalence of gestational hypertension or thrombosis were found when comparing vaccinated and unvaccinated pregnant women [95]. In addition, there was no substantial negative influence on the incidence of premature birth, endometrial rupture, or unexpected ICU hospitalization among vaccinated expecting mothers when it came to delivery [96]. Together, these findings suggest that the administered vaccines might have limited secondary effects on pregnant woman. This information could potentially change as we expect to have more data about the long-term effect of COVID-19 vaccines in the upcoming few years.

## 7. Conclusions

Studies about COVID-19 are increasing exponentially, whereby several researchers are eager to unravel unanswered questions regarding fertility, pregnancy, and fetal outcome. COVID-19 is described as just a mild-to-moderate condition in some articles, but as a severe disease in others. The negative impacts of SARS-CoV-2 infection and associated RAS dysfunction were evident in both male and female fertility; however, whether the effect is due to direct effect of the virus or a consequence of the inflammatory state of the patient is still debated. The virus has also shown an increased morbidity among pregnant patients. This group was especially susceptible to respiratory and coagulation problems. Considering maternal physiological changes during pregnancy with the SARS-CoV-2 infectious process will help researchers better understand the potential consequences for both the mother and the fetus. ACE2, a major key player of RAS, is a critical component of the pathophysiology of SARS-CoV-2 in the reproductive system and has garnered significant attention. The vertical transmission of COVID-19 is not yet established and remains an important question to be answered. In addition, the safety and effectiveness of COVID-19 vaccination on pregnant women and their fetus need further investigation. To answer all these questions, larger studies are still required, and global united efforts are needed to collect reliable and large-scale data. In fact, these speculations need to be supported by evidence-based studies. To this end, multicentric retrospective studies and/or cohort studies could be conducted to investigate the possible consequences of SARS-CoV-2 infection on fertility, pregnancy, and neonatal life.

## Figures and Tables

**Figure 1 biomedicines-10-01775-f001:**
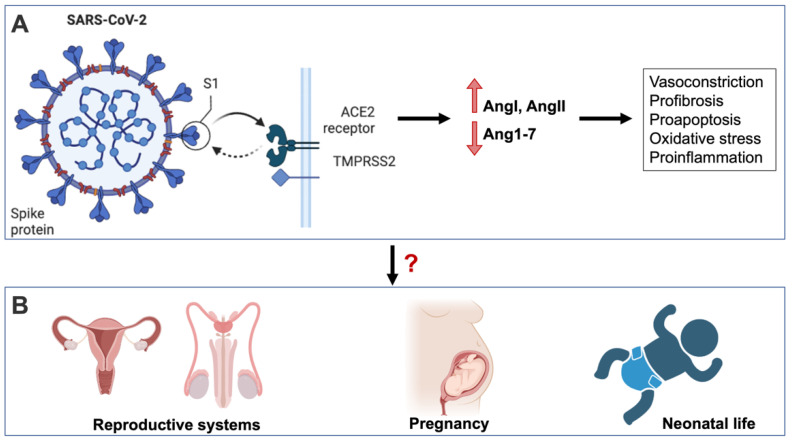
SARS-CoV-2 interaction with ACE2 receptor and impairment of RAS leads to deleterious effects (vasoconstriction, profibrosis, proapoptosis, oxidative stress, proinflammation, proangiogenesis, prothrombosis, and prohypertrophy) in different biological systems (**A**) and potentially procreation (**B**).

**Figure 2 biomedicines-10-01775-f002:**
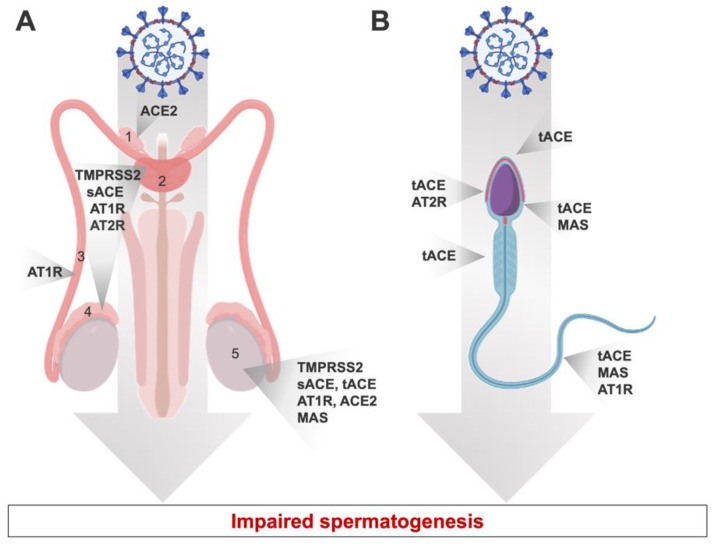
Expression of RAS components in different localizations of the human male reproductive system (**A**) and throughout spermatozoa (**B**). tACE: testicular angiotensin converting enzyme; AT1R: angiotensin II type 1 receptor; AT2R: angiotensin II type 2 receptor; MAS: Mas receptor; TMPRSS2: transmembrane protease serine 2; ACE: angiotensin-converting enzyme; sACE: somatic ACE. 1: vas deferens; 2: prostate; 3: epididymis; 4: seminal plasma; 5: Testis.

**Figure 3 biomedicines-10-01775-f003:**
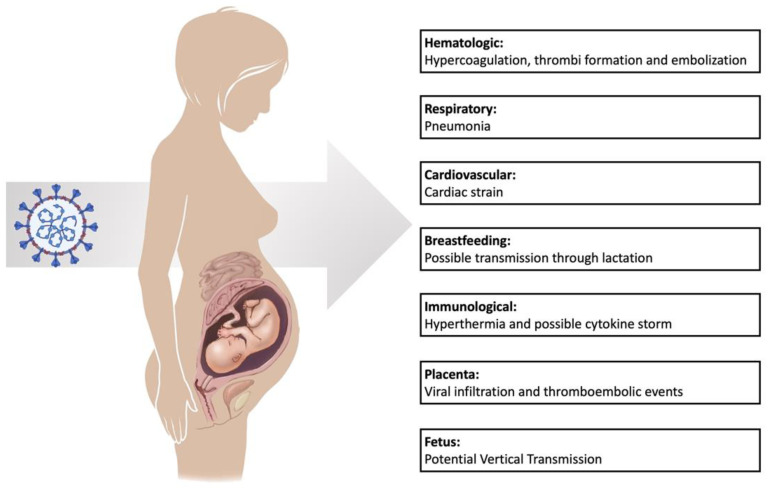
The effects of SARS-CoV-2 on pregnant women.

## Data Availability

Not applicable.

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
