# Peer review of "SARS-CoV-2, COVID-19, and Reproduction: Effects on Fertility, Pregnancy, and Neonatal Life"

_biomedicines, 2022, doi:10.3390/biomedicines10081775_

Round 1

Reviewer 1 Report

In this review, authors summarized the literature reporting the effects of COVID-19 on male and female reproductive systems. Moreover, they evaluated whether the viral infection could impact pregnancy and the effects of COVID-19 vaccines. 

This is a very interesting and well written review that summarizes some important knoledge regarding COVID-19 and its complications. To my opinion the manuscipt only needs minor revision. In particular: 

Line 85: It deserves to be pointed out that, since ACE-2 is expressed on endothelial cells, SARS-CoV2 infection of these cells can lead to endothelial disfunction and inflammation affecting male sexuality (inducing erectile dysfunction) in addition to male fertility (as recently reviewed PMID: 35114008).

Author Response

Reviewer 1:

In this review, authors summarized the literature reporting the effects of COVID-19 on male and female reproductive systems. Moreover, they evaluated whether the viral infection could impact pregnancy and the effects of COVID-19 vaccines. 

This is a very interesting and well written review that summarizes some important knoledge regarding COVID-19 and its complications. To my opinion the manuscipt only needs minor revision. In particular: 

Line 85: It deserves to be pointed out that, since ACE-2 is expressed on endothelial cells, SARS-CoV2 infection of these cells can lead to endothelial disfunction and inflammation affecting male sexuality (inducing erectile dysfunction) in addition to male fertility (as recently reviewed PMID: 35114008).

We thank the reviewer for their comments. We added the idea in the text.

Reviewer 2 Report

Thank you for this informative  study

My comment as follows:

In this review, the authors stated that they compiled the results of articles on the effects of COVID-19 on the male and female reproductive systems. In this context, they stated that they discussed whether the COVID-19 infection affects male and female fertility.

In the introduction of this study, the short history and epidemiological features of COVID-19 infection should be summarized in a sentence or two. In this regard, I recommend that the authors benefit from the following articles:

Sahin TT, Akbulut S, Yilmaz S. COVID-19 pandemic: Its impact on liver disease and liver transplantation. WorldJ Gastroenterol. 2020;26(22):2987-2999. doi: 10.3748/wjg.v26.i22.2987.

BaÅŸkıran A, Akbulut S, Åžahin TT, Tunçer A, Kaplan K, Bayındır Y, Yılmaz S. Coronavirus Precautions: Experience of High Volume Liver Transplant Institute. Turk J Gastroenterol. 2022;33(2):145-152. doi: 10.5152/tjg.2022.21748.

Ganesh B, Rajakumar T, Malathi M, Manikandan N, Nagaraj J, Santhakumar A, Elangovan A, Malik YS. Epidemiology and pathobiology of SARS-CoV-2 (COVID-19) in comparison with SARS, MERS: An updated overview of current knowledge and future perspectives. Clin Epidemiol Glob Health. 2021 Apr-Jun;10:100694. doi: 10.1016/j.cegh.2020.100694.

In this study, the authors prepared a narrative study on the relationship between SARS-Cov-2 virus and infertility and concluded that the disruptions in reproductive physiology in this process may be directly or indirectly related to SARS-Cov-2. however, these speculations need to be supported by evidence-based studies.

Demonstrating an epidemiological relationship between an agent and a disease should either be demonstrated with multicentric retrospective studies or it is possible with cohort studies free from indirect factors.

However, cohort work will never yield the expected result, as the COVID-19 infection appears suddenly and the process is short. In this case, the most appropriate study is retrospective cohort or case-control studies with good multicenter records.

I suggest that the authors point this out as a limitation.

Author Response

Reviewer 2:

My comment as follows:

In this review, the authors stated that they compiled the results of articles on the effects of COVID-19 on the male and female reproductive systems. In this context, they stated that they discussed whether the COVID-19 infection affects male and female fertility.

In the introduction of this study, the short history and epidemiological features of COVID-19 infection should be summarized in a sentence or two. In this regard, I recommend that the authors benefit from the following articles:

Sahin TT, Akbulut S, Yilmaz S. COVID-19 pandemic: Its impact on liver disease and liver transplantation. WorldJ Gastroenterol. 2020;26(22):2987-2999. doi: 10.3748/wjg.v26.i22.2987.

BaÅŸkıran A, Akbulut S, Åžahin TT, Tunçer A, Kaplan K, Bayındır Y, Yılmaz S. Coronavirus Precautions: Experience of High Volume Liver Transplant Institute. Turk J Gastroenterol. 2022;33(2):145-152. doi: 10.5152/tjg.2022.21748.

Ganesh B, Rajakumar T, Malathi M, Manikandan N, Nagaraj J, Santhakumar A, Elangovan A, Malik YS. Epidemiology and pathobiology of SARS-CoV-2 (COVID-19) in comparison with SARS, MERS: An updated overview of current knowledge and future perspectives. Clin Epidemiol Glob Health. 2021 Apr-Jun;10:100694. doi: 10.1016/j.cegh.2020.100694.

As requested, few lines about COVID-19 infection have been added in the introduction using the suggested references.

In this study, the authors prepared a narrative study on the relationship between SARS-Cov-2 virus and infertility and concluded that the disruptions in reproductive physiology in this process may be directly or indirectly related to SARS-Cov-2. however, these speculations need to be supported by evidence-based studies.

Demonstrating an epidemiological relationship between an agent and a disease should either be demonstrated with multicentric retrospective studies or it is possible with cohort studies free from indirect factors.

However, cohort work will never yield the expected result, as the COVID-19 infection appears suddenly and the process is short. In this case, the most appropriate study is retrospective cohort or case-control studies with good multicenter records.

I suggest that the authors point this out as a limitation.

We thank the reviewer for these perspectives. We added them in the text.

Round 2

Reviewer 2 Report

Thank you for this paper

I think the changes made in the manuscript text are sufficient